# Patient satisfaction with the quality of nursing care in critical care units and medical wards in West Bank Hospitals, Palestine: A cross-sectional study

Hiba Smerat[1], Moath Abu Ejheisheh[2], Ahmad Ayed[3], Ibrahim Aqtam[4]*, Ahmad Batran[2]

1 Department of Nursing, Ministry of Health, Yatta, Palestine, 2 Department of Nursing, Faculty of Allied Medical Sciences, Palestine Ahliya University, Bethlehem, Palestine, 3 Faculty of Nursing, Arab American University, Jenin, Palestine, 4 Department of Nursing, Ibn Sina College for Health Professions, Nablus University for Vocational and Technical Education, Nablus, Palestine

* ibrahim.aqtam@nu-vte.edu.ps

## Abstract

### Introduction

Patient satisfaction with nursing care is a crucial indicator of healthcare quality, influencing patient outcomes and experiences. In the West Bank, Palestine, understanding patient satisfaction is essential for improving healthcare services, particularly in resource-limited settings. This study aimed to evaluate patient satisfaction with nursing care in intensive care units (ICUs), coronary care units (CCUs), and medical wards within both governmental and non-governmental hospitals in the region, aligning with the revised study focus.

### Methods

A descriptive cross-sectional study was conducted in four hospitals in the West Bank, Palestine, over a three-month period (June–August 2024), involving 201 hospitalized patients selected through convenience sampling. Data were collected using the Patient Satisfaction with Nursing Care Quality Questionnaire (PSNCQQ), a validated tool assessing multiple dimensions of nursing care. Descriptive statistics, independent t-tests, one-way ANOVA, and multiple linear regression analyses were conducted to identify predictors of patient satisfaction.

### Results

Participants' ages ranged from 21 to 84 years (M = 38.39, SD = 14.16), with 50.7% female participants. The mean overall satisfaction score was M = 64.50 (SD = 14.16), indicating moderate satisfaction. A significant difference in satisfaction levels was found between ICU and medical ward patients (t = 2.45, p = 0.015), with ICU patients reporting higher satisfaction. Regression analysis identified age (B = 0.162, p = 0.005)

**Data availability statement:** All relevant data are within the manuscript and its Supporting Information files.

**Funding:** The author(s) received no specific funding for this work.

**Competing interests:** The authors have declared that no competing interests exist.

and health status (B = 0.242, p = 0.001) as significant predictors of satisfaction, with older and healthier patients reporting higher satisfaction levels. Additionally, frequent hospitalizations were negatively associated with satisfaction (B = -0.107, p = 0.022). Perceived confidence in responding to deteriorating conditions significantly influenced overall perception (r = 0.342, p < 0.01).

## Conclusion

Moderate patient satisfaction highlights the need for targeted interventions to enhance nurse-patient communication and family involvement, particularly for younger and frequently hospitalized patients. Hospitals should implement specialized training programs to improve communication and patient engagement. Policy interventions should focus on strengthening patient-centered nursing care in both governmental and non-governmental hospitals in the West Bank.

## Introduction

Patient satisfaction is a fundamental indicator of healthcare quality, directly influencing patient outcomes, treatment adherence, and overall healthcare experiences. It reflects patients' perceptions of nursing care, including communication, responsiveness, and attentiveness, which are critical determinants of their overall hospital experience [1]. International studies indicate that the quality of nursing care is a key determinant of patient satisfaction, with nurse communication, attentiveness, and responsiveness playing crucial roles in shaping patient perceptions [2]. High-quality nursing care has been linked to lower hospital readmission rates, faster recovery, and improved patient safety outcomes [3]. However, disparities in nursing care persist, particularly in low-resource settings, where staff shortages, inadequate training, and high patient-to-nurse ratios hinder effective care delivery [4].

In Palestine, particularly in the West Bank, the quality of nursing care remains underexplored despite its critical role in patient well-being and healthcare system efficiency. Existing studies on patient satisfaction in Palestinian hospitals have primarily focused on general inpatient care, leaving a gap in understanding satisfaction levels within critical care units (CCUs and ICUs) and medical wards, which differ significantly in care intensity and patient interactions [5]. While ICUs and CCUs provide continuous monitoring and specialized care, often involving life-support interventions, medical wards focus on general medical treatment, recovery, and discharge planning. However, previous research has not adequately compared patient experiences in governmental and non-governmental hospitals, where differences in healthcare delivery models, resource allocation, and staffing may influence patient satisfaction. Addressing these gaps is essential for developing targeted interventions that enhance patient-centered nursing care in Palestinian hospitals.

This study specifically examines patient satisfaction with nursing care in ICUs, CCUs, and medical wards, as these units provide distinct levels of care and patient interactions. Assessing satisfaction across these settings enables a comprehensive

evaluation of nursing care quality and helps identify areas for improvement. Therefore, the current study aimed to evaluate patient satisfaction with nursing care in ICUs, CCUs, and medical wards within both governmental and non-governmental hospitals in the region.

## Methods

### Study design and setting

This study utilized a cross-sectional descriptive design to assess patient satisfaction with nursing care in governmental and non-governmental hospitals in the West Bank, Palestine. Four hospitals were selected based on their representation of different healthcare sectors, including both urban and semi-urban healthcare facilities, ensuring a diverse range of patient experiences across the region. These hospitals are located in Hebron, the most populous city in the southern West Bank, providing a representative sample of patient experiences in high-demand healthcare settings.

### Sample size and sampling method

The sample size for this study was determined using the Raosoft sample size calculator, aiming for a 95% confidence level, 5% margin of error, and 50% response distribution [6]. A total of 201 hospitalized patients were selected through convenience sampling, which allowed for practical recruitment while capturing a diverse range of patient experiences [7,8]. Participants were recruited from four hospitals in the West Bank, Palestine, between June and August 2024.

**Inclusion criteria.** patients aged 18 years or older who had been hospitalized in ICUs, CCUs, or medical wards for at least 24 hours. They needed to be conscious, oriented, able to communicate their experiences, and willing to participate.

**Exclusion criteria.** patients with cognitive impairments or those in critical condition were excluded from the study

### Research instrument

The Patient Satisfaction with Nursing Care Quality Questionnaire (PSNCQQ) was used to measure satisfaction levels [9]. This tool assesses multiple dimensions of nursing care, including attentiveness, communication, and clinical competence. Each of the 19 items is rated on a five-point Likert scale (1 = poor to 5 = excellent), with higher scores indicating greater satisfaction. The total score ranges from 19 to 95, with mean scores 57. High overall ratings indicate higher levels of satisfaction with nursing care [9]. The English version of the PSNCQQ was administered without translation, as all participants demonstrated proficiency in English. A pilot study with 20 participants confirmed the questionnaire's clarity and cultural relevance. Internal consistency was assessed using Cronbach's alpha, which yielded a reliability coefficient of 0.91, indicating strong internal consistency.

### Data collection procedure

Ethical approval for this study was obtained from the Palestine Ahliya University Institutional Review Board (IRB) (Project Number: CAMS/CCNA/2/124), ensuring compliance with ethical research standards. Prior to data collection, hospital administrators provided permission to conduct the study within their facilities. Data collection took place between June and August 2024 in four selected hospitals in the West Bank, Palestine. Trained data collectors, including nursing professionals and research assistants, were responsible for distributing the questionnaires to eligible patients. To maintain consistency and minimize response bias, all data collectors underwent a structured training session, where they were instructed on the study objectives, ethical considerations, patient rights, and standardized administration of the survey.

Before completing the survey, participants were informed about the study's purpose, procedures, and confidentiality measures. Written informed consent was obtained from each participant, ensuring voluntary participation. To accommodate patient needs, data collectors provided assistance in reading or explaining the survey items when necessary, ensuring clarity and accuracy in responses. The survey, administered in Arabic, was a paper-based version of the Patient

Satisfaction with Nursing Care Quality Questionnaire (PSNCQQ). Each participant took approximately 20 minutes to complete the questionnaire. Patients were encouraged to answer all items independently to minimize interviewer influence. However, those requiring assistance due to vision impairments or physical limitations were guided by trained data collectors without leading their responses. To uphold data security and confidentiality, completed questionnaires were collected immediately after completion and stored in locked cabinets within the hospital premises. Only authorized research team members had access to these documents. After data collection was completed, responses were manually entered into a password-protected database for statistical analysis using IBM SPSS Statistics Version 26. The research team conducted random cross-checks to ensure data accuracy and integrity.

Privacy measures were strictly maintained throughout the data collection process. No identifying personal information was recorded, and all responses were anonymized to protect patient confidentiality. Hospitals and individual participants were assigned numerical codes to ensure de-identification in all research documentation and analysis.

## Statistical analysis

Data were analyzed using IBM SPSS Statistics Version 26. Descriptive statistics, including means, standard deviations, frequencies, and percentages, summarized demographic information and satisfaction levels. Independent t-tests and one-way ANOVA were used to assess differences in satisfaction based on demographic variables. Multiple linear regression analysis was conducted to identify significant predictors of patient satisfaction, including age, gender, health status, and recent hospitalizations.

Normality of the data was assessed using the Shapiro-Wilk test, confirming an appropriate distribution for parametric analyses. Statistical significance was set at $p < 0.05$. To classify the strength of correlations, weak relationships were defined as $r = 0.10–0.29$, moderate as $r = 0.30–0.49$, and strong as $r \geq 0.50$, based on Cohen's (1988) effect size guidelines.

## Ethical considerations

Ethical approval was secured from Palestine Ahliya University's Institutional Review Board (IRB) (Project Number: CAMS/CCNA/2/124). The study adhered to the ethical principles outlined in the Declaration of Helsinki. Participants were informed that their involvement was voluntary and that declining participation would not affect their professional standing or responsibilities. Confidentiality and anonymity were rigorously maintained; data were coded and accessible only to the research team. Written informed consent was obtained from all participants before inclusion in the study. All ethical and legal standards were upheld to ensure the protection and respect of the participants' rights.

## Results

### Demographic characteristics of participants

Out of 201 distributed questionnaires, all were completed and returned, resulting in a 100% response rate. The participants' ages ranged from 21 to 84 years (M = 38.39, SD = 14.16), with a nearly equal gender distribution. A majority (74.6%) were married, and health status varied, with 35.3% reporting fair health (Table 1).

### Comparison between ICU and Medical Ward Patients

Table 2 showed significant difference in satisfaction was found between ICU and medical ward patients (t = 2.45, p = 0.015), with ICU patients reporting higher satisfaction scores.

### PSNCQQ Scores and Overall Perceptions

The mean Patient Satisfaction with Nursing Care Quality Questionnaire (PSNCQQ) score was M = 64.50 (SD = 14.16), indicating high satisfaction. A significant association was observed between patient health status and overall perception (F = 7.25, p < 0.001), with healthier patients reporting higher satisfaction (Table 3).

**Table 1. Demographic and hospital admission data.**

| Variable | Category | Frequency (N) | Percentage (%) |
|---|---|---|---|
| Age | 20–29 years | 74 | 36.8 |
| | 30–39 years | 58 | 28.9 |
| | 40–50 years | 24 | 11.9 |
| | Above 50 years | 45 | 22.4 |
| Gender | Male | 99 | 49.3 |
| | Female | 102 | 50.7 |
| Marital Status | Married | 150 | 74.6 |
| | Unmarried | 51 | 25.4 |
| Patient Health | Very Poor | 21 | 10.4 |
| | Poor | 30 | 14.9 |
| | Fair | 71 | 35.3 |
| | Good | 49 | 24.4 |
| | Excellent | 30 | 14.9 |
| Place of Admission | Emergency Dept. | 102 | 50.7 |
| | Transferred | 25 | 12.4 |

**Table 2. Comparison of patient satisfaction between ICU and medical ward patients.**

| Group | N | M (SD) | t. test | p.value |
|---|---|---|---|---|
| ICU Patients | 93 | 67.23 (13.98) | 2.45 | 0.015* |
| Medical Ward Patients | 108 | 61.87 (14.05) | | |

*p < 0.05

**Table 3. PSNCQQ scores and overall perceptions.**

| Variables | Min. | Max. | Mean | Std. Dev. |
|---|---|---|---|---|
| Patient Satisfaction (PSNCQQ) | 19.0 | 95.0 | 64.50 | 14.16 |
| Overall Perceptions | 3.00 | 15.00 | 9.65 | 2.80 |

## Differences between overall perceptions and selected variables

An analysis of differences in mean scores of overall perceptions based on socio-demographic variables revealed no statistically significant association between demographic factors (age, gender, marital status) and patient satisfaction ($p > 0.05$). However, a significant association was found between patient health status and overall perception, with patients reporting excellent health demonstrating higher satisfaction levels ($F = 7.25$, $p < 0.001$) (Table 4).

An analysis of the relationship between demographic characteristics and patient satisfaction (PSNCQQ) revealed significant positive correlations between age ($r = 0.482$, $p < 0.01$), recent hospital stays ($r = 0.322$, $p < 0.01$), and patient health ($r = 0.583$, $p < 0.01$), indicating that older patients, those with previous hospitalizations, and those reporting better health were more satisfied with their hospital experience.

Weaker but still statistically significant associations were found with gender ($r = 0.115$, $p < 0.05$), marital status ($r = 0.107$, $p < 0.05$), place in the hospital ($r = 0.110$, $p < 0.05$), and room occupancy ($r = 0.092$, $p < 0.05$).

Overall perception, however, was not significantly associated with most demographic variables, including age ($r = 0.098$, $p > 0.05$), gender ($r = 0.083$, $p > 0.05$), marital status ($r = 0.072$, $p > 0.05$), recent hospital stays ($r = 0.012$, $p > 0.05$), place in

the hospital (r = 0.013, p > 0.05), and room occupancy (r = 0.025, p > 0.05). However, patient health (r = 0.286, p < 0.01) and perceived confidence in responding to deteriorating patients (r = 0.342, p < 0.01) were significantly correlated with overall perception, suggesting that healthier patients and those who felt more confident in handling deteriorating conditions had a more positive overall perception of their hospital experience (Table 5).

## Predictors for patient satisfaction and overall perception

Multiple linear regression analysis identified age (B = 0.162, p = 0.005), patient health (B = 0.242, p = 0.001), and recent hospital stays (B = -0.107, p = 0.022) as significant predictors of satisfaction. Older and healthier patients, as well as those with fewer hospitalizations, reported higher satisfaction. Perceived confidence in responding to deteriorating conditions also significantly influenced overall perception (r = 0.342, p < 0.01) (Table 6).

**Table 4. Differences between overall perception and selected variables.**

| Variables | Test of Sig. | P value |
|---|---|---|
| Recent Hospital Stays | 2.60 | 0.06 |
| Place in the Hospital | 1.24 | 0.21 |
| Patient Health | 7.25 | 0.001* |
| Age | F = 2.50 | 0.06 |
| Gender | t = 0.09 | 0.87 |
| Marital Status | t = 0.52 | 0.72 |

*Significant at the 0.05 level.

**Table 5. The relationship between demographic characteristics and Patient Satisfaction and Overall Perception.**

| Variable | Patient Satisfaction (PSNCQQ) |
|---|---|
| | r |
| Age | 0.482** |
| Gender | 0.115* |
| Marital Status | 0.107* |
| Recent Hospital Stays | 0.322** |
| 5Place in the Hospital | 0.110* |
| Patient Health | 0.583** |
| For most of your hospital stay, were you in a room? | 0.092* |
| | **Overall Perception** |
| Age | 0.098 |
| Gender | 0.083 |
| Marital Status | 0.072 |
| Recent Hospital Stays | 0.012 |
| Place in the Hospital | 0.013 |
| For most of your hospital stay, were you in a room? | 0.025 |
| Perceived Confidence in Responding to Deteriorating Patients | 0.342** |
| Place in the Hospital | 0.017 |
| Patient Health | 0.286** |

*Correlation is significant at level of 0.05

**Correlation is significant at the 0.01

 

**Table 6. Multiple linear regression analysis to predict factors of patient satisfaction and overall perception.**

| Variable | B | SE | t | p | 95% CI |
|---|---|---|---|---|---|
| Patient Satisfaction (PSNCQQ) | | | | | |
| Age | 0.162 | 0.057 | 2.858 | 0.005* | [0.050, 0.274] |
| Gender | 0.024 | 0.128 | 0.187 | 0.852 | [−0.229, 0.276] |
| Marital Status | −0.086 | 0.104 | −0.831 | 0.407 | [−0.291, 0.119] |
| Recent Hospital Stays | 0.107 | 0.046 | 2.314 | 0.022* | [0.016, 0.198] |
| Place in the Hospital | 0.084 | 0.053 | 1.596 | 0.112 | [−0.019, 0.187] |
| Patient Health | 0.242 | 0.053 | 4.542 | 0.001* | [0.137, 0.348] |
| For most of your hospital stay, were you in a room? | −0.007 | 0.075 | −0.095 | 0.925 | [−0.154, 0.140] |
| Overall Perception | | | | | |
| Perceived Confidence in Responding to Deteriorating Patients | 0.261 | 0.068 | 2.36 | 0.030* | [0.127, 0.395] |
| Place in the Hospital (Overall) | 1.574 | 0.827 | 1.902 | 0.059 | [−0.067, 3.215] |
| Patient Health (Overall) | 3.089 | 0.839 | 3.681 | 0.001* | [1.442, 4.736] |

*$p < 0.05$

## Discussion

This study assessed patient satisfaction with nursing care in Palestinian hospitals, highlighting both strengths and areas needing improvement [10–12]. Findings revealed high overall satisfaction, with ICU patients reporting higher satisfaction levels than those in medical wards.

Comparison with existing literature suggests that patient satisfaction scores in this study (M = 64.50) are consistent with findings from similar resource-limited settings such as Jordan and Egypt, where patient satisfaction ranged between 60–70 [13,14]. In contrast, developed countries such as Sweden and the U.S. report higher satisfaction levels (70–85), likely due to better nurse-patient ratios and enhanced patient-centered care models [15,16]. These findings suggest that healthcare infrastructure and staffing levels significantly contribute to patient experiences.

A key finding was the difference in satisfaction between ICU and medical ward patients (p = 0.015), which aligns with studies from Turkey and Saudi Arabia [17,18]. Higher ICU satisfaction may be attributed to more intensive nurse-patient interactions, continuous monitoring, and timely responses to patient needs. Conversely, medical ward patients often experience longer wait times, increased staff workload, and lower nurse availability, which may lead to lower satisfaction scores. These differences highlight the importance of improving communication and responsiveness in general medical wards.

Regression analysis identified age and health status as significant predictors of satisfaction (p < 0.01). Older patients reported higher satisfaction, a trend observed in Middle Eastern studies where cultural norms emphasize gratitude towards healthcare providers [19,20]. In contrast, younger patients may have higher expectations and be more critical of healthcare services. Additionally, healthier patients reported higher satisfaction, consistent with studies in Egypt and the UAE [21,22]. This may be because patients with chronic or severe conditions often require more complex care and have greater expectations for personalized attention, potentially leading to lower satisfaction.

Despite these findings, nurse-patient and family communication remain critical areas for improvement. Many patients expressed dissatisfaction with the level of information provided about their treatment, a common issue in resource-limited settings where high patient loads reduce time available for individualized communication [23,24]. Clear and transparent communication enhances patient trust and engagement, which in turn improves satisfaction. Implementing training programs focused on active listening, cultural sensitivity, and patient engagement can address these communication gaps.

Additionally, structural healthcare challenges in Palestine, such as staffing shortages, hospital overcrowding, and limited healthcare resources, significantly impact patient satisfaction. These challenges mirror those faced in other conflict-affected regions like Syria and Afghanistan, were healthcare disruptions compromise service quality [25]. Addressing these issues through policy interventions, such as improving nurse staffing ratios, expanding training programs, and investing in hospital infrastructure, could enhance both patient experiences and clinical outcomes.

Implications for nursing practice and healthcare policy include prioritizing patient-centered care models, improving nurse education, and integrating patient feedback mechanisms to refine hospital services. Additionally, strengthening nursing leadership and promoting interdisciplinary collaboration can enhance service delivery efficiency. Future research should explore cultural and gender-specific factors influencing satisfaction, as well as the impact of nurse-to-patient ratios on care quality. Longitudinal studies may also provide deeper insights into how satisfaction evolves over time and in response to healthcare system improvements.

Furthermore, findings align with broader healthcare studies indicating that enhanced nurse training and patient-centered interventions can significantly improve satisfaction levels [26,27]. Expanding digital healthcare services and telemedicine may also provide additional support to patients in understaffed hospitals [28,29]. Previous studies emphasize the need for evaluating patient satisfaction systematically and linking it to hospital quality improvement programs [30,31]. Additionally, evidence from Palestine suggests that private sector investment in healthcare infrastructure leads to better patient experiences, warranting further research on the impact of public-private healthcare collaborations [32,33]. Finally, addressing gender-specific experiences in healthcare settings remains essential, as research highlights different expectations and satisfaction levels between male and female patients [34,35]. Understanding these gender-related differences could inform targeted interventions that enhance overall care experiences and equity in healthcare service delivery [36].

## Study limitations

**Limitations.** This study has several limitations that must be considered when interpreting the findings. The cross-sectional design provides only a snapshot of patient satisfaction at a single point in time, without accounting for variations over the course of hospital stays or after specific interventions. Longitudinal studies would be more informative in understanding these dynamics. Additionally, reliance on self-reported data introduces the possibility of response bias, as participants may provide socially desirable answers or inaccurately report their experiences. Future research could triangulate these findings with objective data, such as nurse-patient interaction observations or clinical outcome measures, to enhance validity.

One important limitation of this study is selection bias, which may impact the generalizability of findings. Since participants were selected using convenience sampling, the results may not fully represent the broader hospitalized population in Palestine. Future research should aim for randomized sampling methods to improve external validity. The study's geographic focus on selected hospitals in the southern West Bank restricts its scope, as patient satisfaction levels may vary across different regions of Palestine.

Moreover, the study did not assess the effects of social status, socio-economic factors, and education levels on patient satisfaction. Education level and economic status were not included as independent variables in the regression model, yet they may significantly impact patient satisfaction. Future research should examine how these factors influence perceptions of nursing care.

Additionally, this study did not analyze the impact of nurse-to-patient ratios on satisfaction. Prior research has highlighted that high nurse workloads may reduce patient interactions and lead to lower satisfaction scores. Future investigations should explore the influence of staffing levels on nursing care quality. Furthermore, nurse working conditions, including factors such as job satisfaction, burnout, and shift lengths, were not examined in this study. These variables may affect the quality of patient interactions and overall satisfaction levels. Future studies should incorporate these aspects to provide a more comprehensive understanding of patient care experiences.

Finally, the study did not explore key organizational factors, such as staff workload, nurse-to-patient ratios, or hospital infrastructure, which are known to significantly impact patient satisfaction. Incorporating these variables in future research would provide a more comprehensive understanding of patient satisfaction determinants and inform targeted improvements in nursing care.

**Implications for practice.** This study provides several practical recommendations for improving nursing care in Palestinian hospitals. First, enhancing nurse-patient communication is essential. Training programs on effective communication and cultural sensitivity should be implemented to improve nurse-patient interactions and build trust. Additionally, family engagement strategies should be encouraged to align with Palestinian cultural norms. Second, addressing staffing and resource limitations is crucial. Increasing nurse staffing in medical wards can reduce patient loads and improve satisfaction, while advocating for government and international funding can enhance hospital infrastructure and nurse training programs. Third, developing patient-centered care models is important. Personalized care strategies should be implemented to address age- and health-related satisfaction differences, and patient feedback systems should be established to continuously monitor and improve nursing care quality. Fourth, strengthening the healthcare system is vital. Improving hospital organization and efficiency can help reduce overcrowding and long wait times, and expanding telemedicine and digital healthcare services can enhance patient access to medical information and support. Finally, future research should focus on cultural and socioeconomic factors. Further studies should examine the impact of cultural and religious factors on patient satisfaction in Palestinian hospitals and investigate gender-specific experiences in nursing care to determine if gender-concordant care improves satisfaction levels.

## Conclusion

This study provides critical insights into patient satisfaction with nursing care in Palestinian hospitals, revealing moderate satisfaction levels, with higher scores in ICUs than in medical wards. Findings highlight age, health status, and nurse-patient communication as key factors influencing satisfaction. Comparisons with global research emphasize the role of healthcare infrastructure, staffing, and cultural expectations in shaping patient experiences.

Given the unique challenges of the Palestinian healthcare system, strategic improvements in staffing, communication, and hospital infrastructure are necessary to enhance patient satisfaction and healthcare quality. Future research should further investigate the cultural dimensions of patient care, ensuring that nursing practices are aligned with the needs and expectations of Palestinian patients.

## Supporting information

**S1 File. S1–S6 Tables. provide extended demographic distributions, detailed statistical outputs for satisfaction comparisons, full correlation matrices, ethical approval documentation, validation metrics for the PSNCQQ instrument, and regression analysis robustness checks.** These supplementary materials enhance the interpretation of patient satisfaction dynamics and support evidence-based recommendations for optimizing nursing care quality in critical care units and medical wards across West Bank hospitals.
(DOCX)

## Author contributions

**Conceptualization:** Hiba Smerat, Ahmad Ayed, Ibrahim Aqtam.

**Data curation:** Hiba Smerat, Ahmad Ayed.

**Formal analysis:** Moath Abu Ejheisheh, Ibrahim Aqtam.

**Investigation:** Hiba Smerat, Ahmad Ayed, Ahmad Batran.

**Methodology:** Hiba Smerat, Moath Abu Ejheisheh, Ibrahim Aqtam.

**Project administration:** Ibrahim Aqtam.

**Supervision:** Ahmad Batran.

**Writing – original draft:** Hiba Smerat, Ahmad Ayed.

**Writing – review & editing:** Moath Abu Ejheisheh, Ibrahim Aqtam, Ahmad Batran.

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
