## [Decision Letter · Decision Letter 0]

20 Feb 2025

PONE-D-25-01316Patient Satisfaction of the Quality of Nursing Care in Intensive Care Units and Medical Wards in South of West Bank Hospital, Palestine: Quantitative StudyPLOS ONE

Dear Dr. aqtam,

Thank you for submitting your manuscript to PLOS ONE. After careful consideration, we feel that it has merit but does not fully meet PLOS ONE’s publication criteria as it currently stands. Therefore, we invite you to submit a revised version of the manuscript that addresses the points raised during the review process.

reply my comments which has been sent by email previously.reply the reviewer feedback

We look forward to receiving your revised manuscript.

Kind regards,

Fadwa Alhalaiqa

Academic Editor

PLOS ONE

Journal Requirements:

2. We note that your Data Availability Statement is currently as follows: “All relevant data are within the manuscript and in Supporting Information files.”

Reviewers' comments:

Reviewer's Responses to Questions

**Comments to the Author**

1. Is the manuscript technically sound, and do the data support the conclusions?

Reviewer #1: Partly

Reviewer #2: Yes

2. Has the statistical analysis been performed appropriately and rigorously? 

Reviewer #1: No

Reviewer #2: Yes

3. Have the authors made all data underlying the findings in their manuscript fully available?

Reviewer #1: Yes

Reviewer #2: Yes

4. Is the manuscript presented in an intelligible fashion and written in standard English?

Reviewer #1: No

Reviewer #2: Yes

5. Review Comments to the Author

Reviewer #1: Dear Authors

After reading the manuscript, I have the following comments:

- This paper needs editing.

- The tiltle needs to be modified. No need to mention south of Palestine.

- Abstract: Generally, it needs restucturing and organization. you used abbreviations such as ICU, CCU only once

thus you need to mention full words. Purpose should be clearly identified (you mentioned inside the manuscript

that you selected governmental and nongovernmental hospitals while in abstract you mentioned only

nongovernmental hospitals). Methods: it is better to mention the statistical tests instead of SPSS. Results: you

need to write results in consice and clear way. Conclusion: it should reflect the recommendations based on

significantresults.

- Introduction:You need to discuss indepth the problem concerning the quality of nursing care from patients'

perspective internationally and locally, gaps in the study, and rationale of the study. Also, explain why you selected

ICU and medical wards because there is a difference in nursing care between these areas. Clarify the purpose of

the study.

- Methods: Design: no need to mention quantitative. Also, not necessary to mention the advantagess of the design.

Be consistent in setting of the study. In calculating sample size, mention all components. What about inclusion and

exclusion criteria? Explain more about the tool concerning scoring system. Did you use Arabic version of the tool? if

yes add that and mention validity and reliability. Not necessary to mention not previously validated in Palestine,

you can calculate the Chronbach's alpha on your study sample. Concerning data anlysis, you did analysis for

predictors using linear regression but you did not mention that in statistical analysis, also you did not assess

normality for the sample.Explain in detaiils about data collection procedure.

- Results: Just mention the significant results. My suggestion concerning difference in sociodemographic to delete

the results and table. My suggestion if you want to do comparison between patients in ICU nad medical wards

according to QONC. Also, I would like to explain that in any regression test you need to apply the assumptions of

regression, thus, apply these assumptions such as normality, correlation,........

- Discussion: It should be indepth in discussion. Compare your results with other countries. Considering the

healthcare system in Palestine. You discuused the influence of cultural factors but you didn't examine these factors.

- Add implications for practice.

Reviewer #2: Dear Author;

How the sample was selected.The effects of social status, socio-economic factors and education levels on the study results were not mentioned.Does level of education affect the degree of satisfaction?Does economic situation affect the degree of satisfaction?Does the number of nurses per patient affect satisfaction?Although it is stated in the limitations, its impact should be briefly mentioned..These questions and the working conditions of nurses should be added to the analysis.What is the current situation of the country and its impact on health policies?

6. PLOS authors have the option to publish the peer review history of their article (what does this mean? ). If published, this will include your full peer review and any attached files.

**Do you want your identity to be public for this peer review?** For information about this choice, including consent withdrawal, please see our Privacy Policy .

Reviewer #1: No

Reviewer #2: No

---

## [Author Response · Author response to Decision Letter 1]

21 Feb 2025

Thanks for valuable feedback and comments, please to see the response to editor an reviewer file

---

## [Decision Letter · Decision Letter 1]

21 Mar 2025

PONE-D-25-01316R1Patient Satisfaction with the Quality of Nursing Care in Intensive Care Units and Medical Wards in West Bank Hospitals, Palestine: A Quantitative StudyPLOS ONE

Dear Dr. aqtam,

Thank you for submitting your manuscript to PLOS ONE. After careful consideration, we feel that it has merit but does not fully meet PLOS ONE’s publication criteria as it currently stands. Therefore, we invite you to submit a revised version of the manuscript that addresses the points raised during the review process. Be sure to:

Address the reviewer feedback 

We look forward to receiving your revised manuscript.

Kind regards,

Fadwa Alhalaiqa

Academic Editor

PLOS ONE

Journal Requirements:

Reviewers' comments:

Reviewer's Responses to Questions

**Comments to the Author**

1. If the authors have adequately addressed your comments raised in a previous round of review and you feel that this manuscript is now acceptable for publication, you may indicate that here to bypass the “Comments to the Author” section, enter your conflict of interest statement in the “Confidential to Editor” section, and submit your "Accept" recommendation.

Reviewer #1: (No Response)

Reviewer #2: All comments have been addressed

2. Is the manuscript technically sound, and do the data support the conclusions?

Reviewer #1: Yes

Reviewer #2: Yes

3. Has the statistical analysis been performed appropriately and rigorously? 

Reviewer #1: Yes

Reviewer #2: Yes

4. Have the authors made all data underlying the findings in their manuscript fully available?

Reviewer #1: Yes

Reviewer #2: Yes

5. Is the manuscript presented in an intelligible fashion and written in standard English?

Reviewer #1: No

Reviewer #2: Yes

6. Review Comments to the Author

Reviewer #1: Dear Authors

After reading the manuscript, I have the following comments:

- This paper still needs editing. Are you sure form abbreviation of medical ward. It should be

consistency in using quality of nursing care in all manuscript. When you mention the abbreviations for the first time no need to repeat them in all manuscript.

- The title: The authors need to mention the type of design such as Cross-sectional Design. Also, the

study included ICU, CCU, and medical nurses, thus it is better to modify ICU to critical care Units. -Abstract: the authors need to modify the purpose to be consistent with the new suggested title. Add duration of data collection. Also, write results in appropriate way M= , SD= ) no need for range. This statement “The study highlights the need for targeted interventions to enhance nurse-patient communication, particularly for younger and more frequently hospitalized patients”, should be in conclusion. No need to repeat results in conclusion, just focus on implication of the significant results. - Introduction: You need to discuss in-depth the problem concerning the quality of nursing care from patients' perspective internationally and locally. My question is “there is a difference between these units and medical wards, how you can clarify that.

- Methods: What about the setting, you mentioned only four hospitals, thus, clarify the method of selecting hospitals. Are these hospitals representing Palestine? Still you need to explain inclusion and exclusion criteria. Move these statements to limitations “While the convenience sampling method facilitated data collection, it may introduce selection bias and limit the generalizability of findings. Future research should explore random or stratified sampling methods to enhance the representativeness of patient demographics across different hospital settings”. Still you did not explain the scoring and cutoff point for the scale. Did you use Arabic version of the tool or you translated it? Concerning data analysis, you mentioned weak or moderate relationships, how you determine the strength of the relationship.

Results: Just mention the significant results, no need to repeat the results.

- Discussion: It should be in-depth interpretation of the results.

Reviewer #2: The authors are grateful for making the requested corrections.The authors have successfully addressed the reviewer’s concerns, making the study a valuable contribution to the field. The manuscript is nearly ready for publication, with only minor refinements needed.

7. PLOS authors have the option to publish the peer review history of their article (what does this mean? ). If published, this will include your full peer review and any attached files.

**Do you want your identity to be public for this peer review?** For information about this choice, including consent withdrawal, please see our Privacy Policy .

Reviewer #1: No

Reviewer #2: **Yes: ** Associate Professor Semih Aydemir

---

## [Author Response · Author response to Decision Letter 2]

21 Mar 2025

We sincerely appreciate the reviewers’ and editor’s time and insightful suggestions, which have significantly strengthened the manuscript. We hope the revisions meet the journal’s standards and look forward to your favorable decision.

Best regards,

Dr Aqtam

---

## [Editor Report · Decision Letter 2]

28 Mar 2025

Patient Satisfaction with the Quality of Nursing Care in Critical Care Units and Medical Wards in West Bank Hospitals, Palestine: A Cross-sectional Study

PONE-D-25-01316R2

Dear Dr. Ibrahim Aqtam,

We’re pleased to inform you that your manuscript has been judged scientifically suitable for publication and will be formally accepted for publication once it meets all outstanding technical requirements.

Kind regards,

Fadwa Alhalaiqa

Academic Editor

PLOS ONE
---

## [Editor Report · Acceptance letter]

PONE-D-25-01316R2

PLOS ONE

Dear Dr. aqtam,

I'm pleased to inform you that your manuscript has been deemed suitable for publication in PLOS ONE. Congratulations! Your manuscript is now being handed over to our production team.

Kind regards,

on behalf of

Pro Fadwa Alhalaiqa

Academic Editor

PLOS ONE